# The More Realism, the Better? How Does the Realism of AI Customer Service Agents Influence Customer Satisfaction and Repeat Purchase Intention in Service Recovery

**DOI:** 10.3390/bs14121182

**Published:** 2024-12-11

**Authors:** Yuting Hu, Ya Xiao, Yi Hua, Yi Fan, Feng Li

**Affiliations:** 1School of Bussienss, Jiangnan University, Wuxi 214122, China; hyt@jiangnan.edu.cn (Y.H.); 1085200117@stu.jiangnan.edu.cn (Y.X.); 1089200212@stu.jiangnan.edu.cn (Y.F.); 2School of Accounting, Nanjing Audit University, Nanjing 211815, China; mp2202136@stu.nau.edu.cn

**Keywords:** AI customer service agent, realism, consumer satisfaction, repeat purchase intention, service recovery

## Abstract

Nowadays, human customer service is increasingly being replaced by artificial intelligence (AI) customer service agents. Service recovery plays a crucial role in shaping consumer experiences and business profitability. The realism of AI agents can significantly impact users’ attitudes and behaviors. However, it remains unclear how different types of realism in AI customer service agents affect customers during service recovery. Drawing on social response theory and expectation–confirmation theory, this study explores the impact of AI agents’ realism on consumer satisfaction and repeat purchase behavior during service recovery, as well as the underlying mechanisms of this effect. We collected data from 784 participants in three studies. Study 1 results show that form realism and behavioral realism of AI customer service agents affect customer satisfaction in successful service recovery situations. Study 2 indicates that the interaction effect of form realism and behavioral realism only influence satisfaction customer satisfaction and repeat purchase intention in terms of failed service recovery. Study 3 further explores the mechanism of action in failed service recovery, finding that perceived warmth and competence mediate the relationship between realism and satisfaction. We systematically examine the influence of the realism of AI customer service agents on consumer satisfaction across distinct success and failure service recovery scenarios, effectively addressing a critical research void. Additionally, our findings offer valuable insights to business managers, empowering them with actionable strategies for enhancing consumer satisfaction across varied consumption contexts and effectively mitigating the repercussions of consumption failures.

## 1. Introduction

The recent rapid development of artificial intelligence, natural language processing, and big data have prompted innovations in the service industry [1,2]. Among these, AI customer service agents are gaining popularity among enterprises, such as the insurance assistant that serves users on Facebook Messenger and voice-activated virtual assistants (e.g., Alexa, Siri, and Cortana) with persona and voice modes [3].

Service recovery refers to the process of mitigating the impact of a service failure that falls short of customer expectations of customer service [4]. Service providers often use apologies, explanations, refunds, free services, and other methods to achieve service recovery [4]. Failure to achieve effective service recovery can lead to customer dissatisfaction [5] or customer switching behavior [6]. Therefore, service recovery is critical for consumer experiences and business profits. The advantages of AI customer service agents in service recovery are many. First, using technologies such as speech recognition and natural language understanding, AI customer service agents can solve consumers’ issues in a personalized and efficient manner [7,8]. Second, the utilization of AI customer service agents allows companies to reduce customer support costs [9]. Despite these advantages, some consumers are still unconvinced that AI customer service agents can become qualified substitutes for human employees in service recovery [7]. Many AI customer service agents have failed to meet consumer expectations and are on the verge of termination owing to some design-related errors [10,11]. Additionally, only a few studies have offered guidelines for the effective design of AI agents in service recovery [12].

The realism of customer service agents refers to how much a robot resembles a human; it comprises form realism and behavioral realism [12]. Recently, some studies revealed that realism is an indispensable element for AI customer service agents [7,12]. The realism of AI customer service agents can exert multifaceted impacts on consumers and companies in customer service. Studies have revealed that realism can enhance the long-term service quality of AI customer service agents by building a satisfactory consumer–brand relationship [7,13]. High-realism online assistants can ensure that customers have very satisfactory and enjoyable shopping experiences, as well as increase their purchase intentions [14,15,16]. In summary, the realism of AI customer service agents can influence consumers’ attitudes and purchase intentions in customer service. Service recovery is the core part of customer service. Therefore, it is valuable to examine the realism of AI customer service agents in terms of service recovery.

However, determining whether the role of realism is positive or negative is a controversy that has raged unabated in service recovery research [12]. When robots exhibit considerable anthropomorphic appearances and behaviors, the level of affinity with them decreases dramatically, resulting in the “Valley-of-Terror Effect”, during which the customer is disgusted by the AI agents [17]. Additionally, as AI agents are so human-like, consumers interact considerably with them, have high expectations, and discuss topics that are unrelated to consumption, such as in the case of IKEA’s AI agent, Anna. Since realism is not always advantageous, suitable realism designs must be considered to create positive customer experiences and increase economic benefits in service recovery.

AI agents that constitute an indispensable aspect of intelligent customer service are exerting critical influences on consumers’ experiences. Whether the role of realism is positive or negative is a controversy in service recovery. Considering different service recovery situations, will the role of realism change? How does the realism of AI agents influence customers? In order to answer these questions, Study 1 chiefly explores the influence of the realism of AI customer service agents on customer satisfaction in a successful service recovery condition regarding their form realism and behavioral realism. Additionally, previous research has focused on customers’ experiences and emotions [18,19] and not on purchase intentions. As customers’ repeat purchase intentions are of major concern to companies [20], Study 2 offers insights to help companies learn the impact of the realism of AI customer service agents on customer satisfaction and repeat purchase intentions in a failed service recovery condition. Study 3 further explores the mechanism by which the realism of AI agents influences customers.

## 2. Theoretical Issues and Hypotheses Development

### 2.1. Realism of AI Customer Service Agents and Consumer Satisfaction

AI customer service agents, which are new forms of frontline service agents, are employed for consumer service [21]. The taxonomy of the realism of AI agents mainly comprises form and behavioral realism [12]; we discuss the functions of AI agents from these perspectives. Form realism refers to the extent to which AI agents’ appearance resembles humans [22] including facial and bodily appearances [23], as well as other demographic profiles (age, name, gender, or race) [18,24]. Behavioral realism refers to the degree to which agents behave like humans [22].

The literature has shown that social response theory applies to human–computer interactions, where people tend to apply social rules when computers exhibit human-like attributes, such as vocal or kinetic cues [25]. Studies in the field of human–computer interactions revealed that people unconsciously apply social rules, such as politeness and reciprocity, to computers exhibiting social cues (appearances, voices, languages, etc.) [26]. Thus, the more form realism AI agents exhibit, the more likely they are to stimulate human schemas. This forces people to believe that these agents can have rational thoughts and display conscious actions like humans, thus forcing people to develop human-like emotions and enhance reliability toward them [7,27]. Therefore, this study proposes that the form realism of AI customer service agents influences consumer satisfaction.

**H1a.** 
*High (vs. low) form realism corresponds to a higher degree of consumer satisfaction.*


Additionally, AI agents exhibiting high behavioral realism can facilitate natural interactions with consumers and promote consumers’ hedonic and utilitarian benefits, thus ensuring high consumer satisfaction [28]. Compared with non-conversational robot assistants with low behavioral realism, conversational robot advisers are more likely to earn affective trust from users, thus enhancing firm images and gaining more benevolent evaluations for companies [29]. Additionally, the high-level behavioral realism of AI agents can satisfy instrumental or relational needs [30], which are the main purposes of communication, to facilitate consumer engagement and enhance their overall shopping satisfaction.

**H1b.** 
*The high (vs. low) behavioral realism corresponds to a comparatively higher degree of consumer satisfaction.*


Both kinds of realism positively affect consumer satisfaction, and the interaction between form and behavioral realism influences consumer satisfaction. Based on the expectation–confirmation theory [31], a decreasing level of satisfaction will emerge when the outcome is lower than the consumers’ expectation, and this will result in negative disconfirmation that reduces customer satisfaction. However, when the outcome is better than the expectation, a positive disconfirmation will result in increased customer satisfaction [32].

When AI agents with high form realism make low-level errors as opposed to those with low form realism, their high form realism will likely contradict the customers’ expectations [19], thereby negatively influencing consumer satisfaction. Conversely, when AI agents exhibit low form realism and high behavioral realism, consumers do not produce high expectations as obtainable with high behavioral realism [12], which will result in a high level of satisfaction. Therefore, the interaction between both kinds of realism exerts different effects on consumer satisfaction.

**H1c.** 
*Form and behavioral realism interact to influence the degree of consumer satisfaction.*


### 2.2. Satisfaction and Consumer Repeat Purchase Intentions

Numerous studies have confirmed that overall service satisfaction can affect consumers’ switching intentions [33], as well as their eventual repeat purchase intentions [13]. Consumer satisfaction is a critical factor that determines consumer repeat purchase intentions [34]. Customers’ repeat purchase intentions are highly significant for the profitability of e-commerce services [20]. Oliver also indicated that consumer satisfaction affects their purchase intentions [31]. He observed that high levels of satisfaction improved consumers’ attitudes toward a brand and consequently increased their repeat purchase intentions. Therefore, this study proposes the following:

**H2.** 
*Consumers’ satisfaction influences repeat purchase intentions.*


### 2.3. Service Recovery Situation as a Moderator

Service recovery situations can be divided into successful service recovery and failed service recovery [35]. Successful service recovery is the process of mitigating the impact of a service failure that falls short of customer expectations on customers and service [35]. Successful service recovery can enhance customers’ perceptions of the quality of services and the organization and improve customers’ satisfaction [36]. However, as AI agents are in the initial development process in the service industry, problems are inevitable. A failed service recovery situation refers to unsuccessful attempts to address a service failure due to unresolved tensions among customers, employees, and process recovery [35]. Failed service recovery situations can lead to substantial revenue losses [37]. Indeed, both service recovery situations have implications for businesses and consumers. However, different underlying mechanisms may play roles in different situations. In a successful service recovery situation, consumers can maintain timely contact with the agents and resolve their issues smoothly, which will improve the rate of positive experiences [38]. Based on social response theory, the research indicated that the realism of AI agents can generate positive attitudes among customers toward AI agents [7]. But cognitive bias indicates that people tend to exaggerate negative experiences rather than positive ones [39]. Thus, customers may be insensitive to the interaction of the two types of realism in AI agents. In failed service recovery situations, customers may experience negative disconfirmation between form and behavioral realism [19]. Negative disconfirmation occurs when the behavioral realism of AI agents fails to meet customers’ previous expectations based on form realism [12], which negatively impacts customer satisfaction [19]. Therefore, this study proposes that the service recovery situation acts as a moderator between the level of realism and consumer satisfaction.

**H3.** 
*The service recovery situation moderates the relationship between the interaction of the realism of AI chatbots and consumer satisfaction. Compared with the success scenario, form and behavioral realism interact to influence consumer satisfaction in failed service recovery situations.*


### 2.4. The Mediation of Warmth and Competence

According to the social response theory, people perceive robots as social actors and unconsciously apply social cognition to them [27,40]. The form and behavioral realism of AI customer service agents might stimulate human schemas such as social perceptions. The stereotype content model, as documented in the literature, focuses on perceived warmth and competence to capture individuals’ social perceptions of others or groups [41]. Perceived warmth refers to helpfulness, sincerity, trustworthiness, and morality, whereas perceived competence typically includes capability, intelligence, effectiveness, and skillfulness, which an intelligent assistant should demonstrate in problem-solving [42]. Previous studies find that the perceived competence and perceived warmth of robots impact users’ usage intentions [43] and loyalty intentions [44]. Therefore, study 3 elucidated the mediating role of perceived warmth and competence between the realism of AI customer service agents and consumer satisfaction within the context of failure service recovery, which has a potentially severe negative impact on the outcomes.

Based on the form realism of AI agents, consumers form an expectation of the services of such AI agents. The higher the form realism, the higher the expectations [12]. However, the failure of service recovery can readily induce disconfirmation of expectations [31], leading consumers to cast doubt upon the authentic capabilities of AI customer service agents. This, in turn, results in a diminished level of perceived competence attributed to AI agents.

**H4a.** 
*Form realism has a negative influence on the perception of AI customer service agents’ competence.*


Meanwhile, AI agents with high behavioral realism can possess a high level of sensing autonomy, thought autonomy, and action autonomy [44], which leads to the inferences of capability, intelligence, efficiency, and effectiveness. Therefore, we hypothesize the following:

**H4b.** 
*Behavioral realism has a positive influence on the perception of AI customer service agents’ competence.*


Nevertheless, the proposition that heightened form realism consistently engenders an enhanced user perception of competence remains equivocal. This uncertainty stems from users’ propensity to harbor elevated anticipations and exacting requisites concerning AI customer service agents’ functionalities. Consequently, nurturing user contentment and sustaining prolonged engagement with these agents becomes intricate in instances where users discern a misalignment between the agents’ offerings and their specific exigencies or initial anticipations [31]. Individuals commonly assess the form realism of an AI chatbot through its avatar and greeting. Elevated levels of formal realism tend to engender heightened expectations of behavioral realism. However, when faced with failed service recovery, the AI chatbot’s inability to meet consumer needs and address inquiries leads to a situation where the disparity between formal realism and behavioral realism fails to align with initial expectations. This exacerbates the diminishment of consumers’ perceived competence of the robot. Specifically, when AI agents with a high level of form realism display low (vs. high) behavioral realism, the perception of AI agents’ competence will be low. When AI agents with a low level of form realism display high (vs. low) behavioral realism, the perception of AI agents’ competence will be high. Hence, the interplay between these two manifestations of realism yields distinct impacts on the perception of AI agents’ competence.

**H4c.** 
*Form and behavioral realism interact to influence the perception of AI customer service agents’ competence.*


An AI agent characterized by a heightened degree of form realism is identified as offering a multitude of advantages. These include facilitating interactions with users and bolstering their social and emotional engagement [45]. The elevated realism of such agents enables them to emulate human-like attributes, encompassing visual appearances and nomenclature [43]. However, in the failure service contexts, an AI agent with a high level of form realism is still able to encourage consumers to apply social rules to respond [25]. Consequently, users tend to form a spectrum of perceptions akin to those experienced during human interactions [46]. This imparts a lasting impact on users, fostering feelings of amiability, assistance, genuineness, reliability, and ethical integrity [45]. Therefore, it can be inferred that the form realism of the AI agent positively influences the perceived warmth.

**H5a.** 
*Form realism has a positive influence on the perception of AI customer service agents’ warmth.*


Behavioral realism refers to the degree to which agents behave like humans [22]. Behavioral realism can promote in-depth natural interactions with consumers. Han and Yang indicated that it is critical to make user interfaces intimate like a “human” friend to enhance affective perceptions regarding the approachability and benevolence of intelligent personal assistants [47]. These characteristics convey a clear indication of the AI agents’ apparent consideration for its users, who in turn acknowledge the demonstrated effort in portraying friendliness and kindness.

**H5b.** 
*Behavioral realism has a positive influence on the perception of AI customer service agents’ warmth.*


The interaction between form and behavioral realism affects how warm AI customer service agents are perceived. The underlying reason for the interaction between form realism and behavioral realism affecting the perception of warmth in AI agents is that people’s perceptions are shaped by evaluating the interaction between how the agents are presented (form realism) and how they behave (behavioral realism). When AI customer service agents with high form realism exhibit low behavioral realism, people may perceive these agents as less warm and friendly because they fail to communicate and interact in a realistic and natural manner [48]. Conversely, when AI customer service agents with low form realism display high behavioral realism, they are perceived as warmer and friendlier because their behavior is more likely to resonate with human users, fostering a sense of empathy and trust [49]. Specifically, when AI customer service agents with a high level of form realism display low (vs. high) behavioral realism, the perception of AI agents’ warmth will be low. When AI agents with a low level of form realism display high (vs. low) behavioral realism, the perception of AI after-sale agents’ warmth will be high. Thus, we posit the following hypothesis:

**H5c.** 
*Form and behavioral realism interact to influence the perception of AI customer service agents’ warmth.*


Several studies indicated that an AI agent demonstrating high (vs. low) warmth prompted extended engagement and a heightened inclination to collaborate with customers [45,50]. Agents possessing a perceived high level of competence can flexibly adapt their technical prowess to cater to the unique requirements of individuals for specific tasks. This results in a high level of perceived competence and warmth of AI customer service agents, ultimately augmenting the consumer’s overall experience. Therefore, we hypothesize the following:

**H6a.** 
*The perceived competence of AI customer service agents positively influences the level of consumer satisfaction.*


**H6b.** 
*The perceived warmth of AI customer service agents positively influences the level of consumer satisfaction.*


This study constructs the following hypothetical model (Figure 1):

## 3. Study 1

### 3.1. Procedures and Sample

Study 1 was conducted using a two-factor, 2 (form realism: low vs. high) × 2 (behavioral realism: low vs. high), between-subject design to explore the influence of the realism of AI agents on consumer satisfaction to test Hypothesis 1.

The experimental procedure comprised three parts. In the first part, the participants were required to read a passage about a hypothetical situation in which customers asked for return service. In the second part, the participants in one of the four experimental groups were randomly assigned to read different dialogues. Thereafter, they answered some questions according to the instructions.

For this experiment, 200 participants were recruited from the Credamo platform, and the data of 12 participants were excluded because they were completely neutral, had never used AI services, or had taken extremely long or short periods to provide answers. Finally, 188 effective data were left (effectiveness rate = 94%). Among these 188 participants, 66 (35.11%) and 122 (64.89%) were male and female, respectively. The average age of participants was 31.51 years (SD = 7.62, range = 20–60). All the participants had experience using AI agents.

### 3.2. Measures

Based on the extant studies on AI agents [7,12,28], we designed two levels of form realism for AI customer service agents. In the high form realism case, the image of the AI customer service agents exhibited real people’s heads, as well as used first person or human names. The AI agents with low form realism exhibited a head with a facial outline and did not use a first person or human name. To test the effectiveness of the manipulation of form realism, we pretested the experimental materials. In the pretest, 40 participants were required to fill out the questionnaire. Four items, such as “the appearance of the AI agent is similar to that of the real customer service agent” (1 = strongly disagree; 7 = strongly agree), were used to measure the degree of form realism [28]. Cronbach’s α of form realism was 0.90. The paired samples *t*-test revealed that the degree of the low form-realism group (*M* = 3.74, *SD* = 1.39) was significantly lower than that of the high form-realism group (*M* = 5.09, *SD* = 1.01, *t*(39) = −6.36, *p* < 0.001).

Following previous studies, behavioral realism is generally measured by social orientation [51]. Therefore, behavioral realism was measured using six items [52] on a seven-point Likert scale (1 = strongly disagree; 7 = strongly agree). For instance, “The AI customer service agent wants to help me”. Cronbach’s α of behavioral realism in this study was 0.85. The SPSS 25.0 software was employed to conduct the paired samples *t*-test for behavioral realism, and the result indicated that the scores of the low behavioral-realism group (*M* = 4.68, *SD* = 0.98) were lower than that of the high behavioral-realism group (*M* = 5.46, *SD* = 0.89, *t*(39) = −5.46, *p* < 0.001). The manipulation of behavioral realism is generally effective.

Customer satisfaction was measured using six items on a seven-point Likert scale (1 = strongly disagree; 7 = strongly agree) [53]. Customer satisfaction includes satisfaction with the services of the AI agents and online store, such as “I am satisfied with the services of the AI customer service agent” and “Generally speaking, I am satisfied with the service provided by the online store”. Cronbach’s α of satisfaction in this study was 0.81. Moreover, other relevant variables and demographic variables were recorded, including experience using an AI service, income, age, and gender.

### 3.3. Results

#### 3.3.1. Realism Manipulation Check

In the last questionnaire section, the participant completed a realism manipulation check. The independent sample *t*-test of form realism indicated that the low form realism (*M* = 4.45, *SD* = 1.43) was significantly lower than the high form realism (*M* = 5.01, *SD* = 1.23, *t*(187) = −2.90, *p* < 0.01). Additionally, the independent sample *t*-test for behavioral realism revealed that the low behavioral realism (*M* = 5.22, *SD* = 1.04) was significantly lower than the high behavioral realism (*M* = 5.70, *SD* = 0.71, *t*(187) = −3.64, *p* < 0.001). Thus, form and behavioral realism were effectively manipulated.

#### 3.3.2. Main Effect

The analysis results revealed that form realism significantly impacted consumer satisfaction. The satisfaction toward the low form realism (*M* = 5.76, *SD* = 0.06) was lower than that toward the high form realism (*M* = 5.94, *SD* = 0.06, *F* (1,181) = 3.92, *p* < 0.05), supporting Hypothesis H1a. Furthermore, behavioral realism significantly impacted customer satisfaction. The satisfaction toward low behavioral realism (*M* = 5.75, *SD* = 0.06) was lower than that toward high behavioral realism (*M* = 5.96, *SD* = 0.06, *F* (1,181) = 5.63, *p* < 0.05), confirming Hypothesis H1b. However, Study 1 revealed that the interaction between form and behavioral realism did not affect consumer satisfaction (*F* (1,181) = 0.02, *p* > 0.05). Therefore, Hypothesis H1c could not be verified in Study 1.

## 4. Study 2

After analyzing the results of Study 1, we did not observe any significant relationship between the interaction of the two types of realism on satisfaction. However, research has revealed that the interaction between form and behavioral realism in failed-service scenarios negatively affects consumer experience [19] because of cognitive bias, in which people tend to exaggerate the negative outcomes they have experienced rather than positive ones [39]. In a successful service recovery, consumers can maintain timely contact with the AI agents and resolve their issues smoothly. In this case, no significant difference will exist between the satisfaction degree [38]. However, once consumers experience an unsuccessful service recovery, they will nurture a negative impression of AI agents and high dissatisfaction [13]. Moreover, unsuccessful service recovery can lead to substantial revenue losses [37]. Therefore, this study explored the relationship between the realism of AI customer service agents, consumer satisfaction, and repeat purchase intention.

### 4.1. Procedures and Sample

Study 2 was conducted using a three-factor, 2 (form realism: low vs. high) × 2 (behavioral realism: low vs. high) × 2 (service recovery situation: success vs. failure), between-subject design. The participants were randomly assigned to read different dialogues in one of the eight experimental groups. In the failure scenario, the participants were required to judge the attributions of the scenario. Thereafter, they offered their opinions regarding satisfaction, repeat purchase intention, and demographic information.

For this study, 400 participants were recruited from the Credamo platform, and the data of 10 participants were excluded because they were completely neutral, had never used AI services, or offered answers within a very short period. Afterward, 390 effective data were left, with an effectiveness rate of 97.5%. Regarding the genders of the participants, 150 (38.5%) and 240 (61.5%) were male and female, respectively. Their average age was 31.72 years (*SD* = 7.67 and range = 19–59).

### 4.2. Measures

In Study 2, we pretested the accuracy of four settings of the failure scenarios (e.g., “Do you consider this service recovery is unsuccessful?”) (*M* = 5.58, *SD* = 1.41, *t*(30) = 6.25, *p* < 0.001; *M* = 5.77, *SD* = 1.28, *t*(30) = 7.70, *p* < 0.001; *M* = 5.74, *SD* = 1.23, *t*(30) = 7.84, *p* < 0.001; *M* = 5.65, *SD* = 1.40, *t*(30) = 6.53, *p* < 0.001), which indicated that customers considered the service recovery scenario to be an actual failure. Beyond form realism, behavioral realism, and consumer satisfaction, Study 2 tested the repeat purchase intentions via three items; Cronbach’s α of repeat purchase intentions was 0.95. Moreover, the participants attributed errors to the AI agents (*M* = 5.45, *SD* = 1.27, *t*(198) = 16.13, *p* < 0.001).

### 4.3. Manipulation Check

The manipulation check for form realism indicated that low form realism (*M* = 4.11, *SD* = 1.68) was significantly lower than high form realism (*M* = 4.45, *SD* = 1.55, *t*(388) = −2.08, *p* < 0.05). Additionally, the manipulation check for behavioral realism indicated that low behavioral realism (*M* = 4.49, *SD* = 1.43) was significantly lower than high behavioral realism (*M* = 4.90, *SD* = 1.38, *t*(388) = −2.82, *p* < 0.01). The manipulation of form and behavioral realism is effective.

### 4.4. Moderation Role of Service Recovery Situations

The main effect of service recovery situations is significant (*F* (1,379) = 3153.88, *p* < 0.001), i.e., the satisfaction from a successful service recovery (*M* = 5.99, *SD* = 0.05) was higher than that from a failed service recovery (*M* = 2.26, *SD* = 0.05). The interaction among the service scenario, form realism, and behavioral realism was also significant (*F* (1,379) = 4.48, *p* < 0.05) (Figure 2 and Figure 3). Specifically, the results of the simple effect test indicated that under the conditions of failed service recovery, the consumers facing low form realism and high behavioral realism (vs. low behavioral realism) expressed high levels of satisfaction (*M*_low behavioral realism_ = 2.09, *M*_high behavioral realism_ = 2.56, *p* < 0.001). Conversely, when the consumers were faced with failed service recovery, high behavioral realism, and high form realism of the AI agents (vs. low form realism), they exhibited low levels of satisfaction (*M*_high form realism_ = 2.13, *M*_low form realism_ = 2.56, *p* < 0.01). However, under successful service recovery, the effect of the interaction between form and behavioral realism was not significant.

### 4.5. Mediation Role of Satisfaction

In the failed service recovery, the conditional indirect effect analysis (Table 1) revealed that the indirect effect of satisfaction differed at various realism values supporting H2. Regarding the AI customer service agents exhibiting low form realism, their behavioral realism significantly affected customers’ repeat purchase intentions via the mediating effect of satisfaction (β = 0.44, lower limit confidence interval (*LLCI*) = 0.17, upper limit confidence interval (*ULCI*) = 0.75). However, regarding the AI agents with high form realism, the mediating effect of satisfaction was insignificant (β = −0.19, *LLCI* = −0.44, *ULCI* = 0.16). When the AI agents exhibited high behavioral realism, their form realism strongly affected their repeat purchase intentions via the mediating effect of satisfaction (β = −0.41, *LLCI* = −0.73, *ULCI* = −0.12). However, for the AI agents exhibiting low form realism, the mediating effect of satisfaction was insignificant (β = 0.15, *LLCI* = −0.08, *ULCI* = 0.45). In the successful service recovery, the conditional indirect effect of satisfaction was insignificant.

## 5. Study 3

In Study 2, we identify a significant correlation between the interaction of two factors, realism and satisfaction, specifically in failed service recovery. However, arriving at this conclusion merely scratches the surface; Study 3 aspires to unveil the underlying mechanistic processes at play.

### 5.1. Procedures and Sample

Study 3 was conducted using a two-factor, 2 (form realism: low vs. high) × 2 (behavioral realism: low vs. high) between-subject design. The participants were randomly assigned to read different dialogues in one of the four experimental groups. The participants were required to judge the attributions of the scenario. Then, they offered their opinions regarding perceived competence, perceived warmth, satisfaction, and demographic information.

For this study, 210 participants were recruited from the Credamo platform, and the data of four participants were excluded because they were completely neutral or offered answers within a very short period. Afterward, 206 effective data were left, with an effectiveness rate of 98.1%. Regarding the genders of the participants, 81 (39.3%) and 125 (60.7%) were male and female, respectively. Their average age was 31.55 years (*SD* = 7.12 and range = 20–59).

### 5.2. Measures

In study 3, we used the experimental material of Study 2. In addition, this study tested questionnaires of perceived competence, perceived warmth [41], product involvement, and satisfaction. Perceived warmth was measured by the items “When I talk to agents: there is a sense of sincerity/friendliness/human warmth”. Perceived competence was measured by asking “Do you think agents are efficient/powerful/skilled”. Cronbach’s α of perceived competence and perceived warmth were 0.87 and 0.88. In this study, Cronbach’s α of satisfaction was 0.93. In addition, this study tested involvement [54], which was regarded as a control variable (i.e., Do you think the product is important/of concern for you?). The correlation coefficient r was 0.83.

### 5.3. The Role of Realism in Competence and Warmth

The main effect of form realism on competence was significant (*F* (1,198) = 5.80, *p* < 0.05), i.e., perceived competence from low form realism (*M* = 2.47, *SD* = 0.11) was higher than that from high form realism (*M* = 2.09, *SD* = 0.11), supporting Hypothesis 3a. The main effect of behavioral realism on competence was not significant (*F* (1,198) = 0.70, *p* = 0.40). The interaction between form realism and behavioral realism on competence was marginally significant (*F* (1,198) = 2.92, *p* = 0.09). Hypothesis 3b and 3c were not confirmed.

The main effect of behavioral realism on warmth was significant (*F* (1,198) = 4.44, *p* < 0.05), confirming Hypothesis 4a. To be more specific, perceived warmth from high behavioral realism (*M* = 3.28, *SD* = 0.15) was higher than that from low form realism (*M* = 2.84, *SD* = 0.14). The main effect of form realism on warmth was not significant (*F* (1,198) = 1.38, *p* = 0.24). The interaction between form realism and behavioral realism on warmth was significant (*F* (1,198) = 4.37, *p* < 0.05), supporting Hypothesis 4c. Specifically, the results of the simple effect test indicated that when facing high behavioral realism AI agents, the consumers perceived higher levels of warmth for low form realism AI agents (vs. high form realism)(*M*_low form realism_ = 3.61, *M*_high form realism_ = 2.95, *p* < 0.05). On the other hand, when the consumers were faced with low form realism AI agents with high behavioral realism (vs. low behavioral realism), they exhibited higher warmth (*M*_high behavioral realism_ = 3.61, *M*_low behavioral realism_ = 2.74, *p* < 0.01).

### 5.4. Mediation Role of Warmth and Competence

Regression analysis (Table 2) showed that behavioral realism and the interaction of form realism and behavioral realism significantly influenced perceived warmth (β = 0.56, *p* < 0.01; β = −0.55, *p* < 0.05); at the same time, perceived warmth positively impacted satisfaction (β = 0.12, *p* < 0.01), confirming H5b. When form realism was low, the mediating effect of perceived warmth between behavioral realism and satisfaction was 0.06 (*LLCI* = 0.01, *ULCI* = 0.16). However, facing high form realism, the mediating effect of perceived warmth between behavioral realism and satisfaction is insignificant (*LLCI* = −0.05, *ULCI* = 0.05). Moreover, the interaction of form realism and behavioral realism influenced perceived competence, presenting marginal significance (β = −0.48, *p* = 0.09); at the same time, perceived competence positively impacted satisfaction (β = 0.64, *p* < 0.001). When form realism was high, the mediating effect of perceived competence between behavioral realism and satisfaction was −0.23 (*LLCI* = −0.48, *ULCI* = −0.02). When form realism was low, the mediating effect of perceived competence between behavioral realism and satisfaction was insignificant (*LLCI* = −0.19, *ULCI* = 0.37). To some extent, the results support Hypothesis 5a.

## 6. Discussion

This study examined the impacts of the realism of AI customer service agents on consumer satisfaction under successful or failed service recovery scenarios. Further, it examined why the realism of AI customer service agents influenced consumer satisfaction and repeat purchase intentions through two experimental studies. The results indicated that under the successful service recovery condition, the main effects were significant; however, there was no interaction, and thus the mediation effect was insignificant. Under the failed service recovery, form and behavioral realism interacted to influence customer satisfaction. Further, consumer satisfaction played a mediating role between realism and customers’ repeat purchase intentions. Different levels of realism affected different perceived competence and perceived warmth, which in turn influenced consumer satisfaction. We observed partly empirical support for the hypotheses.

Our results confirmed that the realism of AI customer service agents influenced consumer satisfaction. Further, the social response theory indicated that people perceived robots as social actors and reacted to them accordingly [40]. According to the social response theory, consumers considered AI agents to be members of society, unconsciously applying social rules, such as politeness and reciprocity, in their interactions with them [26]. Therefore, the higher the form and behavioral realism of AI agents, the more they are likely to stimulate human schemas, causing people to believe that they are capable of human-like rational thinking and conscious actions, thereby developing human-like emotions and attitudes toward them [27]. Furthermore, form and behavioral realism significantly impacted consumer satisfaction under the successful or failed service recovery scenarios; the main effects are significant.

This study also revealed that the service scenario moderated the effect of the realism of AI customer service agents on consumer satisfaction. Our findings correlated with those of extant studies [12], which posit that in the failed service recovery context, failures from high-anthropomorphism robots would provoke more negative attitudes from customers compared with those from low-anthropomorphism ones [12,28]. In the after-sales service that was considered in this study, the basic needs of consumers are to have their problems solved. When basic needs are satisfied, consumers will express satisfaction, although the satisfaction level for basic needs can only attain a certain level, which it cannot cross. If basic needs are not satisfied, consumer satisfaction will decrease significantly. In the context of a successful service recovery, consumers’ basic needs are satisfied, and thus the consumers are satisfied, while in the context of a failed service recovery, they will be dissatisfied. Previous studies revealed that consumers are more sensitive to negative results [39,49] than they are to positive ones; thus, they exhibit exaggerated feelings toward negative results and are almost unmoved by positive results.

The interaction between form and behavioral realism can also be explained using the expectation–confirmation theory [31]. In this theory, confirmation is a crucial factor that affects consumer satisfaction. Before interacting with AI customer service agents, consumers form an expectation of the services of such AI agents based on their form realism. Thus, the higher the form realism, the higher the expectations [12]. The perceived quality of service mainly depends on the behavioral realism of the AI agents; thus, the higher the behavioral realism, the higher the perceived quality of the service. In the failed service recovery context, if the expectation exceeds the perceived quality of the service (form realism is higher than behavioral realism), the consumers will express negative disconfirmation; thus, the greater the gap between consumer expectation and service quality, the lower the satisfaction. Conversely, when AI agents exhibit low form realism and high behavioral realism, consumers will likely experience high satisfaction. Therefore, in the failed service recovery context, the interaction between form and behavioral realism is significant.

Our study indicated that the relationship between realism (form and behavioral) and repeat purchase intentions is mediated by consumer satisfaction. Notably, the mediating effect of satisfaction is only significant in the failed service recovery context; it is insignificant in the successful service recovery context. At the service stage, consumers only contact AI agents to solve their purchasing issues. In the successful service recovery context, the problems are resolved, and most consumers consider it deserved. At that point, an increase in form and behavioral realism would not significantly increase consumers’ repeat purchase intentions. Therefore, in the context of successful service recovery, the mediating effect of consumer satisfaction on repeat purchase intentions is insignificant. In the service recovery, customers will be disgusted by the recurring errors of the service personnel, directly expressing dissatisfaction with the service [55]. In the service recovery, the greater the gap between consumer expectations and actual service quality, the lower the consumer satisfaction. In failed service recovery, consumers may be dissatisfied with AI service, and this would sharply reduce their repeat purchase intention. Therefore, in failed service recovery, the mediating effect of consumer satisfaction between realism and repeat purchase intentions is significant.

In study 3, the first conclusion is that in scenarios of failed service recovery, AI customer service agents with higher form realism will generate lower perceived competence, which in turn reduces consumer satisfaction. AI robots with higher form realism can foster higher expectations for service quality. However, before the service interaction, consumers form expectations as comparative referents through form realism, whereby heightened form realism raises the baseline for competence assessment [56]. There is a mismatch between the actual performance of AI agents and consumer expectations of competence. Consequently, consumers may perceive a lower capability of the robot, aligning with this inconsistency. Simultaneously, we have also found that lower perceived capability can result in decreased consumer satisfaction [45]. This elucidates why we find that in failed service recovery, customer service agents who appear more human-like are perceived to be less competent, thereby reducing satisfaction.

Moreover, a previous study found that strong interactive capabilities can regain consumer trust and satisfaction during service failures [57]. In this context, a customer service representative behaves more like a human and demonstrates stronger interactive capabilities, thereby enhancing the level of warmth perception of AI customer service agents. It has been observed that a high level of perceived warmth positively influences behavioral outcomes, such as user satisfaction and continued usage intentions [40]. Hence, the realism of the behaviors of AI agents positively affects the perception of warmth. Lastly, we also discovered that perceived warmth mediates the mechanism of the interplay between the two types of realism and satisfaction. When AI agents with low form realism exhibit high behavioral realism, it may improve the user’s perceived warmth and satisfaction. Customers might set low expectations for AI agents unlike humans; however, agents’ behavior exceeds the expectations, improving the feeling of warmth and satisfaction.

This study makes key contributions to AI customer service research. First, it fills a gap by examining how AI agents’ form and behavioral realism affect consumer satisfaction and repeat purchase intentions in both successful and failed service recovery contexts, using social response theory and expectation–confirmation theory. It highlights the mediating roles of perceived warmth and competence, offering insights into how realism impacts consumer perceptions. The findings also offer practical implications for designing AI agents that optimize customer satisfaction, especially in service recovery scenarios, providing actionable strategies for businesses to enhance consumer experiences across service settings.

This study opens multiple avenues for further research. First, as the sample was drawn from the Credamo platform, there may be limitations in demographic and cultural representativeness. Since all participants had prior experience with AI customer service, future research could focus on individuals without such experience to enhance the generalizability of the findings. Second, while warmth and competence were examined as mediating variables in Study 3, future studies could explore how factors like AI transparency or escalation to a human agent impact perceptions of warmth and competence. Additionally, it would be valuable to assess whether high competence combined with low warmth is more detrimental than the reverse in service contexts. Finally, although this study centers on AI agents in service recovery, future research could expand to other interactions, such as customer inquiries or pre-failure complaint resolution, broadening the scope of AI’s impact in various service scenarios. This study also does not distinguish between specific types of AI agents, such as chatbots versus voice assistants. Future research could examine how these different formats influence customer satisfaction, clarifying the varied effects of distinct AI types on customer experiences. Moreover, real-world AI interactions can be considered to improve external validity.

## 7. Conclusions

(1)Under the successful service recovery condition, the main effects of form realism and behavioral realism were significant; however, there was no interaction. The mediation effect of consumer satisfaction between realism and customers’ repeat purchase intentions was insignificant.(2)Under the failed service recovery, form and behavioral realism interacted to influence customer satisfaction. Further, consumer satisfaction played a mediating role between realism and customers’ repeat purchase intentions.(3)Form realism and behavioral realism affected perceived competence and perceived warmth, which in turn influenced consumer satisfaction. Perceived competence and perceived warmth played a mediating role between realism and customer satisfaction.

## Figures and Tables

**Figure 1 behavsci-14-01182-f001:**
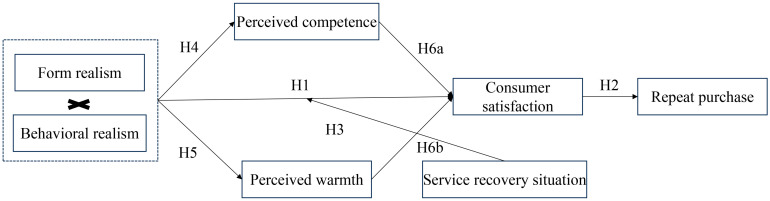
Research model.

**Figure 2 behavsci-14-01182-f002:**
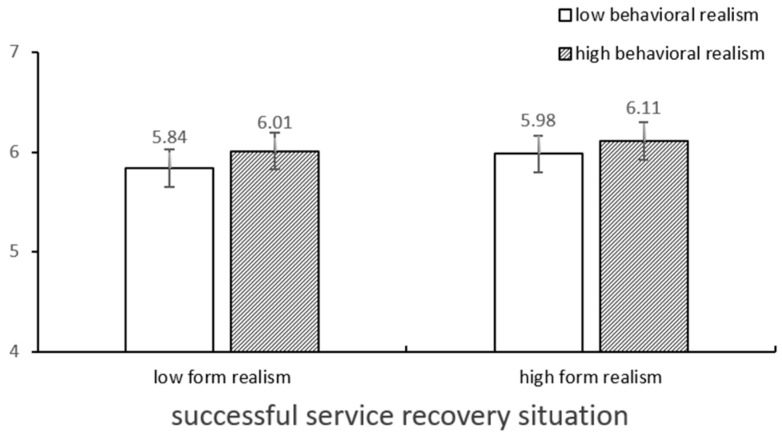
Satisfaction toward types of realism in successful situations.

**Figure 3 behavsci-14-01182-f003:**
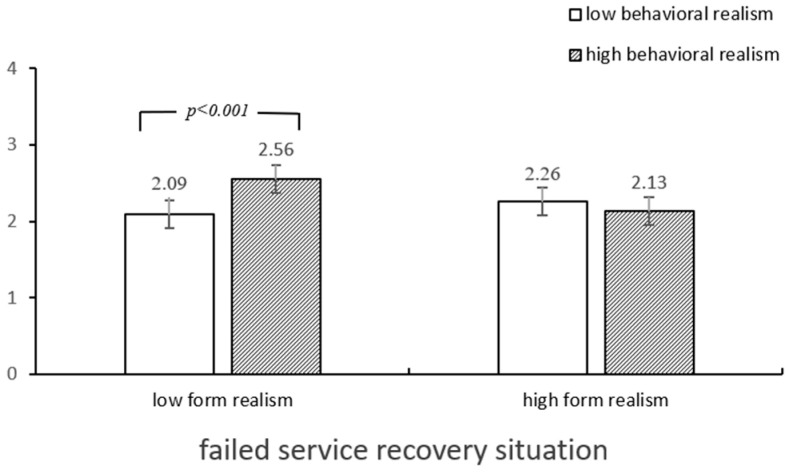
Satisfaction toward types of realism in failure situations.

**Table 1 behavsci-14-01182-t001:** Conditional process analysis of failed service recovery.

	β	*SE*	*t*	*p*	*LLCI*	*ULCI*
Mediator variable model for predicting satisfaction						
Constant	−0.07	0.13	−0.54	0.59	−0.33	0.19
Gender	0.12	0.17	0.70	0.49	−0.22	0.45
Age	0.04	0.08	0.50	0.62	−0.12	0.20
Income	0.15	0.07	2.09	0.04	0.01	0.29
Form realism	−0.17	0.14	−1.24	0.22	−0.44	0.10
Behavioral realism	0.21	0.14	1.49	0.14	−0.07	0.50
Form realism × behavioral realism	−0.73	0.28	−2.65	0.01	−1.28	−0.19
Dependent variable model for predicting repeat purchase						
Constant	−0.01	0.09	−0.15	0.88	−0.18	0.16
Gender	0.02	0.11	0.20	0.84	−0.19	0.23
Age	−0.02	0.04	−0.46	0.64	−0.10	0.06
Income	−0.06	0.05	−1.09	0.28	−0.17	0.05
Form realism	0.01	0.10	0.07	0.94	−0.19	0.21
Behavioral realism	0.10	0.10	0.10	0.33	−0.10	0.30
Form realism × behavioral realism	−0.12	0.19	−0.67	0.51	−0.49	0.24
Satisfaction	0.76	0.06	13.16	*p* < 0.001	0.65	0.88

Note: *N* = 199; bootstrap sample size = 5000; *LL* = lower limit; *CI* = confidence interval; and *UL* = upper limit.

**Table 2 behavsci-14-01182-t002:** Conditional process analysis of the mediation of warmth and competence.

	β	*SE*	*t*	*p*	*LLCI*	*ULCI*
Mediator variable model for predicting warmth						
Constant	−0.30	0.27	−1.14	0.26	−0.82	0.22
Age	0.07	0.07	1.01	0.31	−0.06	0.20
Gender	0.05	0.14	0.35	0.72	−0.22	0.32
Income	0.02	0.07	0.28	0.78	−0.01	0.15
Involvement	0.21	0.07	3.22	*p* < 0.01	0.08	0.35
Form realism	0.12	0.19	0.66	0.51	−0.24	0.49
Behavioral realism	0.56	0.19	2.98	*p* < 0.01	0.19	0.92
Form realism × behavioral realism	−0.55	0.26	−2.09	0.04	−1.08	−0.03
Mediator variable model for predicting competence						
Constant	0.17	0.28	0.61	0.54	−0.38	0.72
Age	−0.01	0.07	−0.12	0.91	−0.15	0.13
Gender	−0.04	0.15	−0.29	0.77	−0.33	0.24
Income	0.05	0.07	0.70	0.49	−0.09	0.19
Involvement	0.03	0.07	0.44	0.66	−0.11	0.17
Form realism	−0.09	0.20	−0.48	0.63	−0.48	0.29
Behavioral realism	0.12	0.20	0.62	0.54	−0.27	0.51
Form realism × behavioral realism	−0.48	0.28	−1.72	0.09	−1.03	0.07
Dependent variable model for predicting satisfaction						
Constant	−0.07	0.14	−0.47	0.64	−0.35	0.21
Age	0.04	0.04	1.06	0.29	−0.03	0.11
Gender	−0.09	0.07	−1.25	0.21	−0.24	0.05
Income	0.11	0.04	3.12	*p* < 0.01	0.04	0.18
Involvement	0.02	0.04	0.58	0.56	−0.05	0.09
Form realism	0.19	0.10	1.97	0.05	−0.0001	0.39
Behavioral realism	0.21	0.10	2.08	0.04	0.01	0.41
Form realism × behavioral realism	−0.29	0.14	−2.07	0.04	−0.58	−0.01
Competence	0.64	0.04	16.08	*p* < 0.001	0.56	0.72
Warmth	0.12	0.04	2.76	*p* < 0.01	0.03	0.20

Note: *N* = 206; bootstrap sample size = 5000; *LL* = lower limit; *CI* = confidence interval; and *UL* = upper limit.

## Data Availability

The dataset is available upon request from the authors.

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
