# Peer review of "The More Realism, the Better? How Does the Realism of AI Customer Service Agents Influence Customer Satisfaction and Repeat Purchase Intention in Service Recovery"

_behavsci, 2024, doi:10.3390/bs14121182_

Round 1

Reviewer 1 Report

Comments and Suggestions for Authors

This research conducted three studies to examine the impact of realism of AI customer service agents on consumer satisfaction and repeat purchase in service recovery and the mechanism of the effect. The research question is interesting and of practical importance. The paper is mostly well-written. Still, there are some issues that should be addressed.

1)The introduction of service failure is a little abrupt. The authors may consider clarifying the relationship between AI customer service, service failure and service recovery more naturally. 

2)The contributions of this paper is not clearly stated. Why is the realism of AI customer service agents important in service recovery?

3) The statement of interaction hypotheses (H1c, H3c, H4c) and the mediation hypotheses H5a and H5b should be more specific for a better understanding. In addition, there is no hypothesis for the moderating effect of service recovery situation, though it appears in the research model.

4)in the method part, the reasons for excluding participants should be specified. For example, in study 1 "took extremely long or short periods to provide answers". What is the criteria for long or short periods? Moreover, what analyses were conducted to test hypotheses in each study should be mentioned.

Good luck with this research!

Author Response

Thank you for the opportunity to revise and resubmit our paper. We appreciate the constructive comments and suggestions that were brought up by you and the reviewers. According to these comments, we made a number of changes to improve our paper, and we believe that addressing these issues has further improved our manuscript.

  1. The introduction of service failure is a little abrupt. The authors may consider clarifying the relationship between AI customer service, service failure and service recovery more naturally.

Response: Thanks for your suggestions. We have amended the statement introducing service failure and service recovery more naturally, such as “Service recovery means the process of mitigating the impact of service failure failing to meet customer expectation on customers and service [4]. Service providers often use apology, explanation, refund, free services and other ways to achieve service recovery[4]. If the business can’t achieve service recovery which can lead to customer dissatisfaction [5] or customer switching behaviour [6]. Therefore, service recovery is critical for consumer experiences and business profits. ”

  1. The contributions of this paper is not clearly stated. Why is the realism of AI customer service agents important in service recovery?

Response:  Thanks for your suggestions. We have added the expression about the importance of the realism of AI customer service agents in service recovery ,such as “The realism of AI customer service agents can exert multifaceted impacts on consumers and companies in customer service. Studies have revealed realism can enhance the long-term service quality of AI customer service agents by building a satisfactory consumer–brand relationship [7,13]. High realism online assistants can ensure that customers have very satisfactory and enjoyable shopping experiences, as well as increase their purchase intentions [14,15,16]. In summary, realism of AI customer service agents can influence consumers’ attitudes and purchase intentions in customer service. Service recovery is the core part of customer service. Therefore, it is valuable to examine the realism of AI customer service agents in service recovery.

However, determining whether the role of realism is positive or negative is a controversy that has raged unabated in service recovery [12]. When robots exhibit considerable anthropomorphic appearances and behavior, the level of affinity with them will decrease dramatically, resulting in the “Valley-of-Terror Effect” during which the customer is disgusted by the AI agents [17]. Additionally, as AI agents are so human-like, consumers interact considerably with them, have high expectations, and discuss topics that are unrelated to consumption, such as IKEA Anna. As realism is not always advantageous, suitable realism designs must be considered to create positive customer experiences and increase economic benefits in service recovery.”

  1. The statement of interaction hypotheses (H1c, H3c, H4c) and the mediation hypotheses H5a and H5b should be more specific for a better understanding. In addition, there is no hypothesis for the moderating effect of service recovery situation, though it appears in the research model.

Response: We have rewritten hypotheses. In addition, we added the hypothesis for the moderating effect of service recovery situation as follows “Service recovery situation can be divided to successful service recovery and failed service recovery [35]. Successful service recovery is the process of mitigating the impact of a service failure that falls short of customer expectations on customers and service [35]. Successful service recovery can enhance customers' perceptions of the quality of services and the organization and improve customers' satisfaction [36]. However, as AI agents is in the initial development process in service industry, problems are inevitable. Failed service recovery situation refers to unsuccessful attempts to address a service failure due to unresolved tensions among customers, employees, and process recovery. [35]. Failed service recovery situation can prompt huge lost in income [37]. Indeed, both service recovery situations have implications for businesses and consumers. But different underlying mechanisms may play roles in different situations. In a successful service recovery situation, consumers can maintain timely contact with the agents and resolve their issues smoothly which will improve positive experience[50][38]. Based on social response theory, the research indicated that realism of AI agents can bring customers positive attitude toward AI agents [7]. But cognitive bias indicates people tend to exaggerate the negative experience rather than positive ones [39]. Thus, customers may be insensitive to the interact of two realisms of AI agents. In failed service recovery situation, customers may experience negative disconfirmation between the form and behavioral realism [19]. Negative disconfirmation occurs when behavioral realism of AI agents fails to meet customer previous expectations based on form realism [12] which negatively impacts customers’ satisfaction [19]. Therefore, this study proposes that the service recovery situation acts as a moderator between the realisms and consumer satisfaction.

  1. In the method part, the reasons for excluding participants should be specified. For example, in study 1 "took extremely long or short periods to provide answers". What is the criteria for long or short periods? Moreover, what analyses were conducted to test hypotheses in each study should be mentioned.

Response: The reasons for excluding participants have be specified,such as study 1 “took extremely short periods to provide answers” study 2 “Six participants had never used AI services. Four participants offered answers within a very short period”. Concrete analyses have been mentioned in the study, such as T-test, ANOVA analysis, regression analysis.

Reviewer 2 Report

Comments and Suggestions for Authors

Journal Name: behavioral sciences

Manuscript Number: behavsci-3286812

Title: The More Realism, the Better? How Does Realism of AI  Customer Service Agents Influence Customer Satisfaction and Repeat Purchase in Service Recovery

I have read with interest the paper entitled “The More Realism, the Better? How Does Realism of AI Customer Service Agents Influence Customer Satisfaction and Repeat Purchase in Service Recovery”. Authors investigate an important aspect related to AI service recovery. The paper has sufficient discussion and a certain degree of originality. However, following suggestions are made to improve the quality of paper.

Comments to the author

Abstract

1. The abstract should point out better what are the prime objectives of this study.

2. Authors should depict a brief explanation about the necessity of carrying out such study? However, the abstract fails to clearly demonstrate concrete empirical evidence by this research.

1. Introduction

The Introduction section requires a significant improvement. Authors should address the following issues and rewrite the this part.

1. There are no research questions clearly delineated and no discussion how author(s) are going to address these research questions. I am sorry but it is quite hard to follow. Also, there is no discussion about how authors are going to address these research questions. This is required, because it helps the reader to understand your research better.

2. The author(s) need to ensure that they are presenting an introduction that clearly draws out the following: what is the problem, where are the theoretical gaps, why do these gaps matter, what have you done and what have you found?

2. Theoretical Issues and Hypotheses Development

(1)  Why is it limited to the context of service recovery, that is, limiting the impact of AI customer service on consumer satisfaction and repeat purchases? Therefore, it is necessary to explain the necessity of applying AI customers in the context of service recovery. The article lacks discussion on the concept and application scenarios of service recovery. It seems that AI customer service is applied in any scenario where users communicate with customers.

(2) I cannot understand what logic H3a's hypothesis is based on, please further argue.

3. Study 1-3

what is the passage about a hypothetical situation, please provide in the text. what’s the questionnaire and scale?

How about the Credamo platform?  

what’s the items of each dimension? please provide the scale as an attachment.

The Cronbach’s α, sd, and mean for variable can be design in a table, this way it looks more readable.

4. Discussion

The author should strengthen the description of research contributions and research prospects.

5.  Literature

Most of the references are from 10 or even 20 years ago, please replace them with references from the past five years.

In summary, I believe that the paper needs minor revisions.

Author Response

Dear Reviewer,

Thank you for the opportunity to revise and resubmit our paper. We appreciate the constructive comments and suggestions that were brought up by you. According to these comments, we made a number of changes to improve our paper, and we believe that addressing these issues has further improved our manuscript.

  1. Abstract

(1) The abstract should point out better what are the prime objectives of this study.

Response: Thanks for your suggestions. The prime objectives of this study have been added in the abstract as follows: “Based on the social response theory and the expectation-confirmation theory, this research revealed the impact of realism of AI customer service agents on consumer satisfaction and repeat purchase in service recovery and the mechanism of the effect. ”

(2) Authors should depict a brief explanation about the necessity of carrying out such study? However, the abstract fails to clearly demonstrate concrete empirical evidence by this research.

Response: We have clarified the necessity of this study as follows: “Nowadays human customer service is gradually replaced by artificial intelligence customer service agents. Service recovery is critical for consumer experiences and business profits. The realism of AI agents would influence users’ attitudes and behaviors. However, we don’t know whether and how different realism of AI customer service agents influences customers in service recovery.”

  1. Introduction

The Introduction section requires a significant improvement. Authors should address the following issues and rewrite this part.

There are no research questions clearly delineated and no discussion of how the author(s) are going to address these research questions. I am sorry, but it is quite hard to follow. Also, there is no discussion about how the authors are going to address these research questions. This is required because it helps the reader to understand your research better.

The author(s) need to ensure that they are presenting an introduction that clearly draws out the following: what is the problem, where are the theoretical gaps, why do these gaps matter, what have you done and what have you found?

Response: In the original manuscript, we have written research questions, but the language is not straightforward enough. We added summary sentences in every paragraph in the introduction. In addition, we rewrote the last paragraph in the introduction as follows: “AI agents that constitute an indispensable aspect of intelligent customer service are exerting critical influences on consumers’ experiences. Whether the role of realism is positive or negative is a controversy in service recovery. Considering different service recovery situations, whether the role of realism change? How does realism of AI agents influence customers? In order to answer these questions, Study 1 chiefly explores the influence of realism of AI customer service agents on customer satisfaction in successful service recovery conditions regarding their form realism and behavior realism. Additionally, previous research has focused on customers’ experiences and emotions [18,19] and not on purchase. As customers’ repeat purchase intentions are of major concern to companies [20], study 2 offers insights to help them learn the impact of realism of AI customer service agents on satisfaction of customers and their repeat purchase intentions in failed service recovery conditions. Study 3 further explores the mechanism by which the realism of AI agents influences customers.

  1. Theoretical Issues and Hypotheses Development

(1)  Why is it limited to the context of service recovery, that is, limiting the impact of AI customer service on consumer satisfaction and repeat purchases? Therefore, it is necessary to explain the necessity of applying AI customers in the context of service recovery. The article lacks discussion on the concept and application scenarios of service recovery. It seems that AI customer service is applied in any scenario where users communicate with customers.

Response:  Thanks for your suggestions. We have added the expression about the necessity of applying AI customers in the context of service recovery and the discussion on the concept and application scenarios of service recovery as follows: “Service recovery means the process of mitigating the impact of service failure failing to meet customer expectation on customers and service [4]. Service providers often use apologies, explanations, refunds, free services, and other ways to achieve service recovery[4]. If the business can’t achieve service recovery, it can lead to customer dissatisfaction [5] or customer switching behavior [6]. Therefore, service recovery is critical for consumer experiences and business profits. There are many advantages of AI customer service agents in service recovery. First, using technologies such as speech recognition and natural language understanding, AI customer service agents can solve consumers’ issues in a personalized and efficient manner [7,8]. Second, the utilization of AI customer service agents allows companies to save customer support costs [9]. Despite these advantages, some consumers are still unconvinced that AI customer service agents can become qualified substitutes for human employees in service recovery [7]. Many AI customer service agents have failed to meet consumer expectations and are on the verge of termination owing to some design-related errors [10,11]. Additionally, only a few researchers have offered guidelines for the effective design of AI agents in service recovery [12].

The realism of customer service agents refers to how much a robot resembles a human; it comprises form realism and behavioral realism [12]. Recently, some researchers revealed that realism is an indispensable element for AI customer service agents [7,12]. The realism of AI customer service agents can exert multifaceted impacts on consumers and companies in customer service. Studies have revealed realism can enhance the long-term service quality of AI customer service agents by building a satisfactory consumer–brand relationship [7,13]. High realism online assistants can ensure that customers have very satisfactory and enjoyable shopping experiences, as well as increase their purchase intentions [14,15,16]. In summary, the realism of AI customer service agents can influence consumers’ attitudes and purchase intentions in customer service. Service recovery is the core part of customer service. Therefore, it is valuable to examine the realism of AI customer service agents in service recovery.

However, determining whether the role of realism is positive or negative is a controversy that has raged unabated in service recovery [12]. When robots exhibit considerable anthropomorphic appearances and behavior, the level of affinity with them will decrease dramatically, resulting in the “Valley-of-Terror Effect,” during which the customer is disgusted by the AI agents [17]. Additionally, as AI agents are so human-like, consumers interact considerably with them, have high expectations, and discuss topics that are unrelated to consumption, such as IKEA Anna. As realism is not always advantageous, suitable realism designs must be considered to create positive customer experiences and increase economic benefits in service recovery.

(2) I cannot understand what logic H3a's hypothesis is based on, please further argue.

Response: There are two paragraphs above Hypothesis 3a. The first paragraph introduced the relevant concepts and theories of mediation variables. In fact, the second paragraph mainly expounds on the relationship between form realism and competence as follows: “Based on their form realism, consumers form an expectation of the services of such AI agents. The higher the form realism, the higher the expectation [12]. However, the failure service recovery can readily induce disconfirmation of expectations [31], leading consumers to cast doubt upon the authentic capabilities of AI customer service agents. This, in turn, results in a diminished level of perceived competence attributed to AI agents.”

  1. Study 1-3

(1) what is the passage about a hypothetical situation, please provide it in the text. What’s the questionnaire and scale?

Response: Thanks for your suggestions. We have attached hypothetical situation pictures in the appendix.

(2) How about the Credamo platform?  

Response: Credamo is a professional research data platform similar to MTurk. It is committed to providing scientific research institutions, enterprises, and individuals with one-stop solutions such as large-scale research/experiment, data collection, modeling analysis, and commercial application. Credamo has 3 million own samples. Independent setting of the compensation for own respondents and automatic payment after review, easily solving the payment problem of respondents. Precise push of questionnaires for specific samples, such as limiting nine major categories of characteristics such as gender, occupation, and region. The platform also supports the collection of international sample data.

Credamo has provided data services for more than 3,000 university teachers and students and more than 4,000 enterprises around the world. The service objects include scholars from top universities such as the Massachusetts Institute of Technology, New York University, the Hong Kong University of Science and Technology, Peking University, Tsinghua University, and Beijing Normal University, as well as leading enterprises in various industries such as China Telecom, Kantar Consulting, Panasonic Electric Appliances, Mars China, Tencent, and Carrefour. The service scope in the academic field covers various disciplines such as management, psychology, medicine, sociology, tourism, and hotel management. The research papers have been accepted by top academic journals (such as PNAS, Psychological Science, Journal of Consumer Research,  American Review of Public Administration, etc.). At the same time, Credamo also provides services such as large-scale research, government affairs management, human resources management, and consumer product development for the government and enterprises. Thus, we think Credamo is credible to use.

(3) what are the items of each dimension? Please provide the scale as an attachment.

Response: Thanks for your suggestions. We have attached scales in the appendix.

(4) The Cronbach’s α, sd, and mean for a variable can be designedin a table, this way, it looks more readable.

Response: In three experiments, Cronbach’s α, sd, and mean for the variable have been reported in the paper. Considering the length and number of words, this paper doesn’t repeat the report. We hope the reviewer can forgive.

   5. Discussion

The author should strengthen the description of research contributions and research prospects.

Response: Due to the word count requirement of the journal, this part was removed from the original manuscript, which has been added as follows:

This study makes key contributions to AI customer service research. First, it fills a gap by examining how AI agents’ form and behavioral realism affect consumer satisfaction and repeat purchase intentions in both successful and failed service recovery contexts, using social response theory and expectation-confirmation theory. It highlights the mediating roles of perceived warmth and competence, offering insights into how realism impacts consumer perceptions. The findings also offer practical implications for designing AI agents that optimize customer satisfaction, especially in service recovery scenarios, providing actionable strategies for businesses to enhance consumer experiences across service settings.

   This study opens multiple avenues for further research. First, given that the sample was drawn from the Credamo platform, there may be limitations regarding demographic and cultural representativeness. Since all participants had prior experience with AI customer service, future research could examine the attitudes and behaviors of individuals without such experience to enhance the generalizability of findings second, while warmth and competence were investigated as mediating variables in Study 3. Future studies could examine how factors like AI transparency or escalation to human agents’ impact perceptions of warmth and competence and assess whether high competence combined with low warmth is more detrimental than the reverse in service contexts. Finally, although this study focuses on AI agents in service recovery, future research could expand to other interactions, such as customer queries or pre-failure complaint resolution, broadening the implications of AI in various service scenarios. This study doesn’t distinguish specific types of AI agents, such as chatbots versus voice assistants. Future research could examine how these different AI formats influence customer satisfaction, helping clarify the varied effects of distinct AI types on customer experiences.

  1.  Literature

Most of the references are from 10 or even 20 years ago; please replace them with references from the past five years.

Response: Thanks for your suggestions. We updated the references. Now, nearly 50% of references were published in the past five years.

Reviewer 3 Report

Comments and Suggestions for Authors

I congratulate you on your interesting and good piece on AI customer service agents. I especially appreciate the clear definitions of every theory/construct and the fact that you performed 3 studies to strengthen your ideas.

As you move forward with the paper, I have a few suggestions you might consider:

1) Please describe the tested scenarios more clearly. You might add an appendix to your paper where you display (at least parts of) the conversations that participants had to read. This is quite important because AI customer service is the focus of your paper but at the moment I cannot read how the AI customer service was designed in your study.

2) Please also display the measurement instruments more clearly. You could add all 3 questionnaires to the appendix or make one big table that shows all constructs and questions (and sources) and that shows for which studies (1, 2 or 3) the constructs were used.

3) There should be a source for each construct, including repeat purchase intention. If you created the items by yourself, you should at least display them.

4) As a reader, It can be quite complicated to follow the 12 hypotheses and 3 studies. Therefore you might present a table that clearly lists all hypotheses, which studies correspond to each hypothesis, and whether it is significant.

5) Another way to reduce complexity would be adding a chapter "Conclusion" that clearly displays and describes your main findings along with managerial implications - like you did it in your abstract.

Comments on the Quality of English Language

I am not a native speaker and I think your English is highly proficient. However, there is certain potential to improve it:

1) There are certain instances where you might add (or delete) an article, such as: (line 136) "Stereotype content model referring ..." or (line 86) Social response theory ..."

2) I'm afraid your sentence on line 67 "Further, as only ..." is not a full sentence.

3) Please also consider certain headlines, such as "2.3 The Mediating of ...", maybe it's "Mediation"?

4) Also consider singular and plural, such as (line 141) "Previous study find that ..."

5) Please reconsider this sentence: (line 307) "Besides, unsuccessful service recovery prompts huge lost income."

6) I think in line 340 you wrote "signature" instead of "significant".

7) I think in line 393 you wrote correlation coefficient instead of Cronbach's Alpha.

Author Response

Dear Reviewer,

Thank you for the opportunity to revise and resubmit our paper. We appreciate the constructive comments and suggestions that were brought up by you and the reviewers. According to these comments, we made a number of changes to improve our paper, and we believe that addressing these issues has further improved our manuscript.

1. Please describe the tested scenarios more clearly. You might add an appendix to your paper where you display (at least parts of) the conversations that participants had to read. This is quite important because AI customer service is the focus of your paper but at the moment I cannot read how the AI customer service was designed in your study.

Response: Thanks for your suggestions. We are sorry for missing the conversations. We have added an appendix to the paper that displaying virtual conversation scenes.

2. Please also display the measurement instruments more clearly. You could add all threequestionnaires to the appendix or make one big table that shows all constructs and questions (and sources) and shows for which studies (1, 2, or 3) the constructs were used.

Response: Thanks for your suggestions. We have attached a table that shows all constructs, questions, and sources in the appendix.

3. There should be a source for each construct, including repeat purchase intention. If you created the items by yourself, you should at least display them.

Response: Thanks for your suggestions. We have written sources on paper. In order to improve readability, we highlighted the source of the questionnaires in the questionnaires table.

4. As a reader, it can be quite complicated to follow the 12 hypotheses and three studies. Therefore, you might present a table that clearly lists all hypotheses, which studies correspond to each hypothesis, and whether it is significant. Another way to reduce complexity would be addinga chapter, "Conclusion," that clearly displays and describes your main findings along with managerial implications - as you did it in your abstract.

Response: Thanks for your suggestions. The results summary in the first paragraph of the discussion may be less striking. We added a chapter, "Conclusion" as follows. Implications and future research directions were mentioned in the discussion.

(1) Under the successful service recovery condition, the main effects of form realism and behavior realism were significant, whereas there was no interaction. The mediation effect of consumer satisfaction between realism and customers’repeat purchase intentions was insignificant.

(2) Under the failed service recovery, the form and behavioral realism interacted to influence customer satisfaction. Further, consumer satisfaction played a mediating role between realism and customers’ repeat purchase intentions.

(3) Form realismand behavior realism affected perceived competence and perceived warmth, which in turn influenced consumer satisfaction. Perceived competence and perceived warmth played a mediating role between realism and customer’ satisfaction.

5. There is certain potential to improve itin language:

Response: Thank you for your detailed modification suggestions.

There are certain instances where you might add (or delete) an article, such as: (line 136) "Stereotype content model referring ..." or (line 86) Social response theory ..."

Response: We amended as “Literature has shown that social response theory applies to human-computer interactions, where people tend to apply social rules when computers exhibit human-like attributes, such as vocal or kinetic cues [25].” “The stereotype content model, as documented in literature, focuses on perceived warmth and competence to capture individuals’ social perceptions of others or groups [36].”

I'm afraid your sentence on line 67,"Further, as only ..." is not a full sentence.

Response: Furthermore, only a few studies have considered successful and failed service recovery situations of AI customer service agents.

Please also consider certain headlines, such as "2.3 The Mediating of ..." maybe it's "Mediation"?

Response: Apologies for the confusion. I have corrected “Mediating” to “Mediation” as suggested.

Also consider singular and plural, such as (line 141) "Previous study find that ..."

Response: A previous study finds that perceived competence and perceived warmth of robots impact users’ usage intention.

Please reconsider this sentence (line 307):"Besides, unsuccessful service recovery prompts huge lost income."

Response: Besides, failed service recovery situations can lead to substantial revenue losses [51].

I think in line 340,you wrote "signature" instead of "significant.”

Response: Apologies for the confusion. I have corrected “signature” to “significant” as suggested.

I think in line 393,you wrote correlation coefficient instead of Cronbach's Alpha.

Response: There are two items, so we used the correlation coefficient.

Reviewer 4 Report

Comments and Suggestions for Authors

This manuscript presents a well-structured experimental design with a thorough literature review, providing valuable insights into how AI customer service agents' realism impacts consumer satisfaction and repeat purchase intentions. The theoretical foundation and research framework are strong, but several areas need further clarification to enhance the study’s contribution and improve result interpretation.

In Study 1, it’s unclear whether participants viewed static AI images or had dynamic interactions. This raises concerns about how the realism experienced compares to real-world AI interactions, which could affect the external validity of the findings.

For customer satisfaction, a 7-point Likert scale was used across studies, but how was it validated for AI-driven service recovery? Were any pre-existing scales adapted or developed specifically for this research? Also, were mediating factors like customer expectations considered, as they might affect satisfaction?

The use of the Credamo platform prompts questions about sample representativeness, particularly across demographics and cultures. This might limit the generalizability of the findings. Since all participants had prior AI customer service experience, how might results differ for those without such experience?

In Study 2, both successful and failed service recovery were examined, but the realism of these scenarios is unclear. Were more complex, ambiguous service failures considered, as seen in real-life settings? This could influence perceptions of AI agents’ warmth and competence. The findings suggest that the form-behavior realism interaction only affects satisfaction in failure, not success. The authors should clarify why this difference occurs and its theoretical implications.

In Study 3, warmth and competence are examined as mediating variables, but the mechanisms by which realism influences these perceptions could be expanded. Could factors like AI transparency or escalation to human agents also impact warmth and competence perceptions? It would also be useful to explore if high competence but low warmth is more detrimental than the reverse.

The study focuses on AI agents in service recovery, but could the findings extend to other interactions, such as customer queries or pre-failure complaint resolution? This would broaden the implications for AI use across service contexts. Finally, discussing the limitations of specific AI agents (e.g., chatbot vs. voice assistant) would help clarify how different AI types affect customer satisfaction.

Lastly, please provide the ethics approval number or details from the institutional review board to confirm ethical compliance in research involving human participants.

Author Response

Dear Reviewer,

Thank you for the opportunity to revise and resubmit our paper. We appreciate the constructive comments and suggestions that were brought up by you and the reviewers. According to these comments, we made a number of changes to improve our paper, and we believe that addressing these issues has further improved our manuscript.

Please note: Our responses are indicated in Blue font.

1. In Study 1, it’s unclear whether participants viewed static AI images or had dynamic interactions. This raises concerns about how the realism experienced compares to real-world AI interactions, which could affect the external validity of the findings.

Response: In three experiments, participants were provided dialog scene pictures in which participants imagined they were the customers in the picture. We have added an appendix to the paper that displaying virtual conversation scenes. Previous studies have used similar experimental manipulation(Crolic et al., 2022; Huang & Dootson,2022; Miao et al., 2022). As the reviewer mentioned, real-world AI interactions can have better external validity. Future research directions suggest applying real-world AI interactions.

2. For customer satisfaction, a 7-point Likert scale was used across studies, but how was it validated for AI-driven service recovery? Were any pre-existing scales adapted or developed specifically for this research? Also, were mediating factors like customer expectations considered, as they might affect satisfaction?

Response: We used pre-existing scales adapted for this research to test satisfaction. The reliability of the questionnaire was tested in this study. Other mediating factors like customer expectations, may exist. We added future studies involving other possible mediating mechanisms, such as customer expectations.

3. The use of the Credamo platform prompts questions about sample representativeness, particularly across demographics and cultures. This might limit the generalizability of the findings. Since all participants had prior AI customer service experience, how might results differ for those without such experience?

Response: (1)Credamo is a professional research data platform that is similar to MTurk. It is committed to providing scientific research institutions, enterprises, and individuals with one-stop solutions such as large-scale research/experiment, data collection, modeling analysis, and commercial application. Credamo has 3 million own samples. Independent setting of the compensation for own respondents and automatic payment after review, easily solving the payment problem of respondents. Precise push of questionnaires for specific samples, such as limiting nine major categories of characteristics such as gender, occupation, and region. The platform also supports the collection of international sample data.

Credamo has provided data services for more than 3,000 university teachers and students and more than 4,000 enterprises around the world. The service objects include scholars from top universities such as the Massachusetts Institute of Technology, New York University, the Hong Kong University of Science and Technology, Peking University, Tsinghua University, and Beijing Normal University, as well as leading enterprises in various industries such as China Telecom, Kantar Consulting, Panasonic Electric Appliances, Mars China, Tencent, and Carrefour. The service scope in the academic field covers various disciplines such as management, psychology, medicine, sociology, tourism, and hotel management. The research papers have been accepted by top academic journals (such as PNAS, Psychological Science, Journal of Consumer Research, American Review of Public Administration, etc.). Thus, we think Credamo is credible to use.

(2) As you mentioned, considering the limitation of the data platform and all participants having prior AI customer service experience, “future research could examine the attitudes and behaviors of individuals from different cultures without such experience to enhance the generalizability of ”

4. In Study 2, both successful and failed service recovery were examined, but the realism of these scenarios is unclear. Were more complex, ambiguous service failures considered, as seen in real-life settings? This could influence perceptions of AI agents’ warmth and competence. The findings suggest that the form-behavior realism interaction only affects satisfaction in failure, not success. The authors should clarify why this difference occurs and its theoretical implications.

Response: Thanks for your suggestions. We have attached hypothetical situation pictures of study 2 in the appendix. Indeed, service failures are complex and ambiguous in real-life settings. One study can’t contain all service failures, but it may be a new research direction. In the discussion, we added future studies can focus on different kinds of service failures that happened in real-life settings.

 The results showed that form-behavior realism interaction only affects satisfaction in failed recovery situations, not success. To clarify the reasons and their theoretical implications, we added expression in the hypothesis and discussion as follows:

“Service recovery situation can be divided into successful service recovery and failed service recovery [35]. Successful service recovery is the process of mitigating the impact of a service failure that falls short of customer expectations on customers and service [35]. Successful service recovery can enhance customers' perceptions of the quality of services and the organization and improve customers' satisfaction [36]. However, as AI agents are in the initial development process in service industry, problems are inevitable. Failed service recovery situation refers to unsuccessful attempts to address a service failure due to unresolved tensions among customers, employees, and process recovery. [35]. Failed service recovery situations can lead to significant revenue losses [51][37]. Indeed, both service recovery situations have implications for businesses and consumers. But different underlying mechanisms may play roles in different situations. In a successful service recovery situation, consumers can maintain timely contact with the agents and resolve their issues smoothly which will improve positive experience [50][38]. Based on social response theory, the research indicated that realism of AI agents can bring customers a positive attitude toward AI agents [7]. But cognitive bias indicates people tend to exaggerate the negative experience rather than positive ones [49][39]. Thus, customers may be insensitive to the interaction of two realisms of AI agents. In failed service recovery situations, customers may experience negative disconfirmation between the form and behavioral realism [19]. Negative disconfirmation occurs when behavioral realism of AI agents fails to meet customers’ previous expectations based on form realism [12], which negatively impacts customers’ satisfaction [19]. Therefore, this study proposes that the service recovery situation acts as a moderator between the realisms and consumer satisfaction.” 

“This study also revealed that the service scenario moderated the effect of the realism of AI customer service agents on consumer satisfaction. Our findings correlated with those of extant studies [12], which posit that in the failed service recovery context, failures from high-anthropomorphism robots would provoke more negative attitudes from customers compared with those from low-anthropomorphism ones [12, 28]. In the service recovery that was considered in this study, the basic needs of consumers are to have their problems solved. When basic needs are satisfied, consumers will express satisfaction, although the satisfaction level for basic needs can only attain a certain level, which it cannot cross. If basic needs are not satisfied, consumer satisfaction will decrease significantly. In the context of a successful service recovery, consumers’ basic needs are satisfied, and they are satisfied, while in the context of a failed service recovery, they will be dissatisfied. Previous studies revealed that consumers are more sensitive to negative results [49] than they are to positive ones; thus, they exhibited exaggerated feelings toward negative results and are almost unmoved by positive results.”

5. In Study 3, warmth and competence are examined as mediating variables, but the mechanisms by which realism influences these perceptions could be expanded. Could factors like AI transparency or escalation to human agents also impact warmth and competence perceptions? It would also be useful to explore if high competence but low warmth is more detrimental than the reverse.

Response: Thanks for your suggestions. AI transparency, escalation to human agents, and high competence but low warmth could be expanded in the future, which has been added to the discussion. For example, “Future studies could examine how factors like AI transparency or escalation to human agents impact perceptions of warmth and competence, and assess whether high competence combined with low warmth is more detrimental than the reverse in service contexts. ”

6. The study focuses on AI agents in service recovery, but could the findings extend to other interactions, such as customer queries or pre-failure complaint resolution? This would broaden the implications for AI use across service contexts. Finally, discussing the limitations of specific AI agents (e.g., chatbot vs. voice assistant) would help clarify how different AI types affect customer satisfaction.

Response: This is a good idea, which has been added to the discussion as follows. “ Finally, although this study focuses on AI agents in service recovery, future research could expand to other interactions, such as customer queries or pre-failure complaint resolution, broadening the implications of AI in various service scenarios. This study doesn’t distinguish specific types of AI agents, such as chatbots versus voice assistants. Future research could examine how these different AI formats influence customer satisfaction, helping clarify the varied effects of distinct AI types on customer experiences.”

7. Lastly, please provide the ethics approval number or details from the institutional review board to confirm ethical compliance in research involving human participants.

Response: Institutional Review Board Statement: Approval was obtained from the ethics committee of Jiangnan University (Approval Code: JNU20230901IRB06). All research was performed following the relevant guidelines of the JN Scientific Research Ethics Committee.

Reviewer 5 Report

Comments and Suggestions for Authors

Comments:

This is a well-written paper containing interesting results which merit publication. The author has made significant contributions to AI customer service agents and service recovery. A few minor revision are list below.

1. The author should propose a hypothesis after H5b, to describe the potential moderating effect of service recovery.

2. In Chapter 4.4, a bar chart or a line chart should be used to present the moderating effect of service recovery.

Author Response

Dear Reviewer,

Thank you for the opportunity to revise and resubmit our paper. We appreciate the constructive comments and suggestions that were brought up by you and the reviewers. According to these comments, we made a number of changes to improve our paper, and we believe that addressing these issues has further improved our manuscript.

Please note: Our responses are indicated in Blue font.

Comments:

1. The author should propose a hypothesis after H5b to describe the potential moderating effect of service recovery.

Response: Thanks for your suggestions. We added the hypothesis for the moderating effect of the service recovery situation as follows: “Service recovery situation can be divided into successful service recovery and failed service recovery [35]. Successful service recovery is the process of mitigating the impact of a service failure that falls short of customer expectations on customers and service [35]. Successful service recovery can enhance customers' perceptions of the quality of services and the organization and improve customers' satisfaction [36]. However, as AI agents are in the initial development process in service industry, problems are inevitable. Failed service recovery situation refers to unsuccessful attempts to address a service failure due to unresolved tensions among customers, employees, and process recovery. [35]. Failed service recovery situations can lead to substantial revenue losses. [37]. Indeed, both service recovery situations have implications for businesses and consumers. However, different underlying mechanisms may play roles in different situations. In a successful service recovery situation, consumers can maintain timely contact with the agents and resolve their issues smoothly, which will improve positive experience [38]. Based on social response theory, the research indicated that realism of AI agents can bring customers a positive attitude toward AI agents [7]. However cognitive bias indicates people tend to exaggerate the negative experience rather than positive ones [39]. Thus, customers may be insensitive to the interaction of two realisms of AI agents. In failed service recovery situations, customers may experience negative disconfirmation between the form and behavioral realism [19]. Negative disconfirmation occurs when behavioral realism of AI agents fails to meet customer previous expectations based on form realism [12], which negatively impacts customers’ satisfaction [19]. Therefore, this study proposes that the service recovery situation acts as a moderator between the realisms and consumer satisfaction.

H3: The service recovery situation moderates the relationship between the interaction of realisms of AI chatbots and consumer satisfaction. Compared with the success scenario, the form, and behavioral realism interact to influence consumer satisfaction in failed service recovery situation.

2. In Chapter 4.4, a bar chart or a line chart should be used to present the moderating effect of service recovery.

Response: Thanks for your suggestions. We added a bar chart presetting the moderating effect of service recovery.

Reviewer 6 Report

Comments and Suggestions for Authors

This study generally addressed the issue of AI-customer service agent, and the authors explored whether both kinds of AI bot realism could affect customer satisfaction and if this could in turn lead to repeat purchase intention. This is a timely topic and an important one for business. However, several issues dampened my enthusiasm. 

1.     The Introduction did not adequately build a case for the focal role of AI realism in the context of service recovery. Normally, for customers seeking customer service online, the most important criteria for them to judge the quality of service should have been “whether it solved my problem”, rather than “if the agent is anthropomorphic enough”. I agree that from a product design perspective, anthropomorphism could exert certain effects, but that’s relatively peripheral compared with “solving the issue”. The authors need to better situate the research question within the literature and the actual transactional context. 

2.     The theoretical model and the hypotheses need a re-organization, In the current form, it is hard to parse the specific hypotheses. Moreover, the “service recovery situation” appeared in Figure 1 but no hypothesis is made about it.

3.     In experiment 1, what kind of dialogue is used (Line 240)? In experiment 2 and 3, a critical variable of this study, the service recovery situation, did not have any details. How were successful or failure situation defined, how were they manipulated? No example dialogue texts provided here. 

4.     The three experiments were all fictional situations, which limited the validity of the study. Participants did not actually engage with the agent, and they were just asked to rate the agent based on a given dialogue text. It would acceptable to rate agent anthropomorphism under this kind of experimental setting, but the results derived from rating service satisfaction as well as repeat purchase intention under this setting would be highly unreliable or unconvincing.

5.     Degrees of freedom for ANOVA result in study 1, t test and ANOVA result in study 2, and ANOVA results in study 3 did not fit the reported sample size of each study. 

6.     Formatting issues exist. E.g., line 412

7.     The manuscript needs English editing. 

Comments on the Quality of English Language

The manuscript needs English editing. 

Author Response

Dear Reviewer,

Thank you for the opportunity to revise and resubmit our paper. We appreciate the constructive comments and suggestions you and the reviewers made. Based on these comments, we made a number of changes to improve our paper, and we believe that addressing these issues has further improved our manuscript.

Please note: Our responses are indicated in Blue font.

Comments:

1. The Introduction did not adequately build a case for the focal role of AI realism in the context of service recovery. Normally, for customers seeking customer service online, the most important criteria for them to judge the quality of service should have been “whether it solved my problem” rather than “if the agent is anthropomorphic enough”. I agree that from a product design perspective, anthropomorphism could exert certain effects, but that’s relatively peripheral compared with “solving the issue.” The authors need to better situate the research question within the literature and the actual transactional context.

Response: Thanks for your suggestions. As you mentioned, “solving the issue” is crucial, but more and more researchers pay much attention to the design of AI services which have been widely used in China. In recent years, papers focusing on related topics have been published in high-quality journals, such as the Journal of Marketing Computers in Human Behavior.

    As you mentioned, the Introduction did not adequately build a case for the focal role of AI realism in the context of service recovery. We revised the introduction completely, increasing the expression of the important role of AI realism in the context of service recovery.

2. The theoretical model and the hypotheses need a re-organization, In the current form, it is hard to parse the specific hypotheses. Moreover, the “service recovery situation” appeared in Figure 1 but no hypothesis is made about it.

Response: Thanks for your reminder. We added the hypothesis 3 as follows. “Service recovery situation can be divided into successful service recovery and failed service recovery [35]. Successful service recovery is mitigating the impact of a service failure that falls short of customer expectations on customers and service [35]. Successful service recovery can enhance customers' perceptions of the quality of services and the organization and improve customers' satisfaction [36]. However, as AI agents are in the initial development process in service industry, problems are inevitable. Failed service recovery situation refers to unsuccessful attempts to address a service failure due to unresolved tensions among customers, employees, and process recovery. [35]. Failed service recovery situations can lead to substantial revenue losses. [37]. Indeed, both service recovery situations have implications for businesses and consumers. But different underlying mechanisms may play roles in different situations. In a successful service recovery situation, consumers can maintain timely contact with the agents and resolve their issues smoothly which will improve positive experience [38]. Based on social response theory, the research indicated that realism of AI agents can bring customers a positive attitude toward AI agents [7]. However, cognitive bias indicates people tend to exaggerate the negative experience rather than positive ones [39]. Thus, customers may be insensitive to the interaction of two realisms of AI agents. In failed service recovery situation, customers may experience negative disconfirmation between the form and behavioral realism [19]. Negative disconfirmation occurs when behavioral realism of AI agents fails to meet customer previous expectations based on form realism [12] which negatively impacts customers’ satisfaction [19]. Therefore, this study proposes that the service recovery situation acts as a moderator between the realisms and consumer satisfaction.

H3: The service recovery situation moderates the relationship between the interaction of realisms of AI chatbots and consumer satisfaction. Compared with the success scenario, the form and behavioral realism interact to influence consumer satisfaction in failed service recovery situations.”

3. In experiment 1, what kind of dialogue is used (Line 240)? In experiment 2 and 3, a critical variable of this study, the service recovery situation, did not have any details. How were successful or failure situation defined, how were they manipulated? No example dialogue texts provided here.

Response: Thanks for your reminder. We have attached hypothetical situation pictures of studies in the appendix. Successful service recovery means the process of mitigating the impact of a service failure that falls short of customer expectations on customers and service. Failed service recovery situation refers to unsuccessful attempts to address a service failure due to unresolved tensions among customers. The specific dialogue texts were added in the appendix.

4. The three experiments were all fictional situations, which limited the validity of the study. Participants did not actually engage with the agent, and they were just asked to rate the agent based on a given dialogue text. It would be acceptable to rate agent anthropomorphism under this kind of experimental setting, but the results derived from rating service satisfaction as well as repeat purchase intention under this setting would be highly unreliable or unconvincing.

Response: As you mentioned, experiment situations may limit the validity of the study. Future research directions suggest applying real-world AI interactions in study. In this study, we used instruction to instruct participants to imagine themselves in the virtual conversations which may improve involving. Previous studies have used similar experimental manipulation(Crolic et al., 2022; Huang & Dootson,2022; Miao et al., 2022).

5. Degrees of freedom for ANOVA result in study 1, t-test and ANOVA result in study 2, and ANOVA results in study 3 did not fit the reported sample size of each study.

Response: Thanks for your reminder. We have checked the results of three studies which were right. The number of control variables in every study may influence freedom.

6. Formatting issues exist. E.g., line 412

Response: Thanks for your reminder. We have changed the information into a subscript format.

7. The manuscript needs English editing.

Response: Thanks for your suggestions. The main content of the study has also been polished by professional institutions. But after language polishing, we changed some paragraphs which may exist some mistakes. We need to polish the full paper more carefully.

Round 2

Reviewer 6 Report

Comments and Suggestions for Authors

The authors have responded to the comments and revised the manuscript accordingly. I have no further comment.